# Performances of Functional and Anatomic Imaging Modalities in *Succinate Dehydrogenase A-*Related Metastatic Pheochromocytoma and Paraganglioma

**DOI:** 10.3390/cancers14163886

**Published:** 2022-08-11

**Authors:** Mayank Patel, Abhishek Jha, Alexander Ling, Clara C. Chen, Corina Millo, Mickey J. M. Kuo, Matthew A. Nazari, Sara Talvacchio, Kailah Charles, Markku Miettinen, Jaydira Del Rivero, Alice P. Chen, Naris Nilubol, Frank I. Lin, Ali Cahid Civelek, David Taïeb, Jorge A. Carrasquillo, Karel Pacak

**Affiliations:** 1Section on Medical Neuroendocrinology, Eunice Kennedy Shriver National Institute of Child Health and Human Development, National Institutes of Health, Bethesda, MD 20814, USA; 2Radiology and Imaging Sciences, Warren Grant Magnuson Clinical Center, National Institutes of Health, Bethesda, MD 20814, USA; 3Nuclear Medicine Department, Radiology and Imaging Sciences, Warren Grant Magnuson Clinical Center, National Institutes of Health, Bethesda, MD 20814, USA; 4Positron Emission Tomography Department, Warren Grant Magnuson Clinical Center, National Institutes of Health, Bethesda, MD 20814, USA; 5Medical Genetics Branch, National Human Genome Research Institute, National Institutes of Health, Bethesda, MD 20892, USA; 6Laboratory of Pathology, National Cancer Institute, NIH, Bethesda, MD 20814, USA; 7Developmental Therapeutics Branch, Center for Cancer Research, National Cancer Institute, National Institutes of Health, Bethesda, MD 20892, USA; 8Early Clinical Trials Development Program, Division of Cancer Treatment and Diagnosis, National Cancer Institute, National Institutes of Health, Bethesda, MD 20892, USA; 9Surgical Oncology Program, Center for Cancer Research, National Cancer Institute, National Institutes of Health, Bethesda, MD 20892, USA; 10Targeted Radionuclide Therapy Section, Molecular Imaging Branch, National Cancer Institute, National Institutes of Health, Bethesda, MD 20892, USA; 11Division of Nuclear Medicine and Molecular Imaging, Department of Radiology and Radiological Science, Johns Hopkins Medicine, Baltimore, MD 21287, USA; 12Department of Nuclear Medicine, La Timone University Hospital, CERIMED, Aix-Marseille University, 13273 Marseille, France; 13Department of Radiology, Molecular Imaging and Therapy Service, Memorial Sloan Kettering Cancer Center, New York, NY 10065, USA

**Keywords:** pheochromocytoma, paraganglioma, metastatic, *SDHA*, PET, CT, MRI, ^68^Ga-DOTATATE, ^18^F-FDG

## Abstract

**Simple Summary:**

Pheochromocytoma and paraganglioma (PPGL) are rare neuroendocrine cancers which carry the risk of metastatic disease. Pathogenic variants in the succinate dehydrogenase subunit A gene *(SDHA)* have been shown to cause metastatic disease, occurring in various regions of the body. Imaging is an early and vital step in the diagnosis and clinical care of these patients. The study here identifies which imaging modality among positron emission tomography (PET), computed tomography (CT), and magnetic resonance imaging (MRI) performs better in localizing metastatic PPGL lesions related to *SDHA*. The study identified that ^68^Ga-DOTATATE PET/CT performed best at overall lesion detection; however, ^18^F-FDG PET/CT performed better in certain anatomic regions of the body. A combined approach with ^68^Ga-DOTATATE and ^18^F-FDG would optimize care and guide clinicians in selecting the appropriate interventions and therapies.

**Abstract:**

The study identifies the importance of positron emission tomographic (PET) and anatomic imaging modalities and their individual performances in detecting succinate dehydrogenase A (*SDHA)*-related metastatic pheochromocytoma and paraganglioma (PPGL). The detection rates of PET modalities—^68^Ga-DOTATATE, ^18^F-FDG, and ^18^F-FDOPA—along with the combination of computed tomography (CT) and magnetic resonance imaging (MRI) are compared in a cohort of 11 patients with metastatic PPGL in the setting of a germline *SDHA* mutation. The imaging detection performances were evaluated at three levels: overall lesions, anatomic regions, and a patient-by-patient basis. ^68^Ga-DOTATATE PET demonstrated a lesion-based detection rate of 88.6% [95% confidence interval (CI), 84.3–92.5%], while ^18^F-FDG, ^18^F-FDOPA, and CT/MRI showed detection rates of 82.9% (CI, 78.0–87.1%), 39.8% (CI, 30.2–50.2%), and 58.2% (CI, 52.0–64.1%), respectively. The study found that ^68^Ga-DOTATATE best detects lesions in a subset of patients with *SDHA-*related metastatic PPGL. However, ^18^F-FDG did detect more lesions in the liver, mediastinum, and abdomen/pelvis anatomic regions, showing the importance of a combined approach using both PET modalities in evaluating *SDHA*-related PPGL.

## 1. Introduction

Succinate dehydrogenase is a critical enzyme participating in cellular aerobic respiration. The enzyme is comprised of four subunits (collectively known as complex II), which converts succinate to fumarate via oxidation. These subunits are represented by the alphabetic characters: A, B, C, and D, where each subunit is expressed by a respective *SDH*-*A*, *B*, *C*, or *D* tumor suppressor gene [1]. Pathogenic variants in these genes can predispose a patient to pheochromocytoma and paraganglioma (PPGL), among other cancers. An international consortium of experts has found that *SDHx*-related pathogenic variants cause disease in 20% of patients with PPGLs [2]. However, patterns of the disease, including clinical manifestations, anatomic location, penetrance, and malignant potential, vary according to the afflicted subunit [3,4]. Identification of these phenotypic patterns, in conjunction with the pathogenic genetic variant in this largely heritable cancer, is critical in the care of patients and their families.

PPGLs are rare neuroendocrine tumors that occur in the adrenal medulla and in sympathetic and parasympathetic ganglia. These tumors can be life-threatening and thus require a thorough clinical workup to prevent morbidity and mortality. Biallelic *SDHA* germline mutations were first identified in a heritable neurodegenerative condition known as Leigh Syndrome [5,6]. Monoallelic pathogenic *SDHA* variants have been associated with tumors such as renal cell carcinoma, gastrointestinal stromal tumor, neuroblastoma, and lung cancers, as well as the focus of this study: PPGLs. The first reported case of *SDHA*-related paraganglioma (PGL) was in 2010 [7]; since then, there have been several studies and observations detailing the potential risk of metastasis [8,9]. While one study using a Bayesian inference predicted a 1.7% penetrance of PPGL for patients with pathogenic variants in *SDHA*, the risk of metastatic disease in those with *SDHA-*related PPGL is up to 66% [10,11].

The clinical picture of metastatic *SDHA*-related PPGL has been reported and observed to be similar in aggressiveness to *SDHB*-related PPGL [8,12]. Since publishing our initial data on ten patients which identified clinical characteristics in metastatic *SDHA*-related PPGL from 2010 to 2018, our group has had one additional patient in this cohort illustrating the rarity of the patients [7]. Although the *SDHA* variants are rare in an already rare condition, the morbidity and mortality related to this genetic predisposition to PPGL are worrisome for patients, their families, and treating clinicians.

In conjunction with biochemical and clinical workup, radiologic imaging pinpoints the anatomic location of PPGLs and whether metastatic disease is present. Imaging can directly translate to therapy (“theranostics”), specifically when functional imaging is correlated to a radiotherapeutic counterpart in patients with inoperable disease. The proposed algorithm from the European Association of Nuclear Medicine/Society of Nuclear Medicine and Molecular Imaging (EANM/SNMMI) in 2019 recommends ^68^Ga-DOTA(0)-Tyr(3)-octreotate (^68^Ga-DOTATATE) as the radiotracer of choice, followed by ^18^F-fluorodeoxyglucose (^18^F-FDG) and ^18^F-fluorodihydroxyphenylalanine (^18^F-FDOPA), in the functional imaging of metastatic PPGL and *SDHx*-related PPGL [13].

In congruence with the EANM/SNMMI guidelines, our study will be the first head-to-head study comparing functional and anatomic imaging modalities in the *SDHA* cohort of PPGL patients. The objective of this study is to extrapolate upon the imaging guidelines by comparing diagnostic performances of ^68^Ga-DOTATATE, ^18^F-FDOPA, ^18^F-FDG, and CT/MRI in *SDHA-*related metastatic PPGL.

## 2. Methods

### 2.1. Selection of Patients

Between 2014 and 2021, 11 consecutive patients (5 female and 6 male) identified with *SDHA*-related metastatic PPGL were prospectively evaluated at the *Eunice Kennedy Shriver* National Institute of Child Health and Human Development (NICHD) at the National Institutes of Health (NIH). All patients had clinically proven metastatic PPGLs based on surgically resected PPGLs, biochemical diagnosis, and anatomic and functional imaging. Patient enrollment into the study began in January 2014 when ^68^Ga-DOTATATE PET/CT was available as a research scan and transitioned to a clinical scan in July 2021 (Clinical Trial Government Identifier: NCT00004847). The protocol was approved by the institutional review board of the *Eunice Kennedy Shriver* NICHD. Informed consent was obtained from all participants and guardians for minors as a part of the investigation.

### 2.2. Patient Cohort and Disease Characteristics

The age of the patients ranged from 16 to 67, with a mean age of 41.9 ± 19.9 years. The mean age at primary PPGL diagnosis was 36.3 ± 18.4 years. The average interval between the first primary PPGL diagnosis and referral to the NIH was 3.3 ± 3.9 years. Nonsense mutations in *SDHA* were most common among these patients, seen in 9 of 11 patients (82%). Eight of 11 patients (73%) had the c.91C>T variant, resulting in a stop gain at codon 31 (p.Arg31*). The other nonsense mutation was seen in patient 3 (c.1534C>T) with a stop gain at codon 512 (p.Arg512*). Full gene deletion was seen in one (patient 8). Patient 6 was the only one with a missense variant (c.1334C>T), changing serine to leucine at codon 445 (p.S445L). Different interpretations of this missense variant are reported in ClinVar as either a variant of uncertain significance (VUS) or a likely pathogenic variant [14]. Additional cohort characteristics, including sex, age when primary PPGL diagnosed, age when imaging scans were performed, location of primary PPGL, biochemical phenotype, time to metastatic disease progression, location of metastatic PPGL, treatments received, Ki-67 percent on histology, and deceased status are summarized in Table 1.

All eleven patients underwent ^68^Ga-DOTATATE, ^18^F-FDG, and either CT or MRI. Additionally, seven patients underwent ^18^F-FDOPA, and six patients underwent both CT and MRI.

### 2.3. Imaging Modality Techniques

Screening CT scans of the neck, chest, abdomen, and pelvis were performed on a multidetector CT device supplied by Siemens (SOMATOM Force and Definition), with the administration of nonionic, low osmolarity iodinated vascular contrast material. Screening MR imaging studies, including the neck, chest, abdomen, and pelvis were performed with either 1.5 or 3.0 Tesla scanners supplied by Philips (Achieva, manufactured Andover, MA, USA) and Siemens (Aera and Verio, manufactured Malvern, PA, USA), using a gadolinium-based contrast agent, and multiple sequences including fat-suppressed and non-fat-suppressed T2-weighted images, in-phase and out-of-phase T1-weighted images, and fat-suppressed sequences following contrast administration.

The PET/CT imaging was performed from the top of the skull to the mid-thighs using time-of-flight and obtained in a 3D mode reconstructed on a matrix suggested by the device manufacturer, Siemens. These scans were performed with a low-dose CT without contrast that was primarily used for anatomical localization. The average lapsed time between imaging and the mean administered radiotracer activity were approximately 60 min and 5.06 mCi for ^68^Ga-DOTATATE, 60.5 min and 7.38 mCi for ^18^F-FDG, and 30.5 min and 12.42 mCi of ^18^F-FDOPA. A 200-mg dose of carbidopa was administered 60 min prior to i.v. ^18^F-FDOPA administration. ^68^Ga-DOTATATE and ^18^F-FDOPA were formulated in the PET Department at the National Institutes of Health as stated in the investigational new drugs application.

### 2.4. Analysis of Data

The CT and MR studies were reviewed by a board-certified diagnostic radiologist (author Ling) with 35 years of experience, including 14 years in PPGL evaluation. All ^68^Ga-DOTATATE, ^18^F-FDG, and ^18^F-DOPA PET/CTs were reviewed by a board-certified nuclear medicine physician with 35 years of experience, including 21 years in PPGL evaluation (author Carrasquillo). Both experts were blinded to clinical information, except for diagnosis, sex, and age. Lesions were considered positive on the PET scans when areas of non-physiologic focal uptake displayed higher maximal standardized uptake values (SUV_max_) than the surrounding tissue.

Imaging studies were performed within a mean duration of 14 ± 19 days of each other. The following analyses were conducted within each imaging study: lesion, region, and patient. In the patient analysis, a patient was determined positive by the presence of a single positive lesion on a particular imaging modality. Similarly, in region analysis, a region was determined positive by the presence of a single positive lesion on a particular imaging modality in the following areas: head and neck, bones, lungs, mediastinum, liver, adrenal glands, and abdomen and pelvis compartments (excluding the liver and adrenal glands). An imaging comparator was constructed from a composite of all the imaging modalities, where a true lesion was deemed to be present if positive on at least two functional modalities or one functional and one anatomic modality. However, if a lesion was only positive on one functional modality or one anatomic modality, the lesion was not considered positive. The composite method was chosen as the reference standard, as a substitute for histologic diagnosis for metastatic lesions, as biopsy and surgical resection for each lesion was not clinically feasible. This imaging comparator was applied in all three analyses: total lesion, anatomic region, and patient-to-patient analyses.

### 2.5. Statistical Analysis

Detection rates are reported as ratios and percentages for lesion detection by each modality with a 95% confidence interval calculated using the GraphPad QuikCalcs website (https://www.graphpad.com/quickcalcs/confInterval1/ accessed on 10 July 2022). McNemar’s tests were performed to compare lesion detection rates between ^68^Ga-DOTATATE and the other functional and anatomic (combined CT/MRI) imaging modalities. A *p*-value < 0.05 shows that a likely difference exists in the detection rates of the modalities being compared. Contingency tables comparing imaging modalities are presented in Appendix A with a *p*-value calculated using GraphPad QuikCalcs website (https://www.graphpad.com/quickcalcs/mcNemar1/ accessed on 10 July 2022).

## 3. Results

### 3.1. Lesion Analysis

Eleven patients with *SDHA-*related PPGL were identified and analyzed for lesions. ^68^Ga-DOTATATE PET/CT had a total lesion detection rate of 88.6% (95% CI, 84.3–92.1), identifying 249 of the 281 composite positive lesions, which was higher than ^18^F-FDG, ^18^F-FDOPA, and CT/MRI (shown in Table 2). Ten primary lesions were found in 8 patients at the time of scanning, and the detection performance on imaging is shown in Table 2. Three patients (Patients 7, 10, and 11) did not have primary lesions imaged at the time of scanning due to resection of these tumors. Three patients (Patients 1, 5, and 8) had imaging of their primary tumors despite interim resections of these tumors due to recurrence of the primary mass. These primary lesions were determined based on tumor anatomical location where chromaffin tissue is physiologically present in conjunction with clinical history. Metastatic lesions were clinically determined based on the following criteria: location in the bones, lungs, or liver; locally invasive recurrent tumors; and presence of tumors where chromaffin tissue is not physiologically present. The detection rate of metastatic lesions by ^68^Ga-DOTATATE outperformed the three other modalities (shown in Table 2).

McNemar’s tests comparing total lesion detection rates found the following two-sided *p*-values: 0.085 for ^68^Ga-DOTATATE vs. ^18^F-FDG, <0.0001 for ^68^Ga-DOTATATE vs. ^18^F-FDOPA, and <0.0001 for ^68^Ga-DOTATATE vs. CT/MRI. The *p*-value less than 0.05 rejects the hypothesis that the modalities’ detection rates are equal, and a difference in performance is evident. Contingency tables comparing lesion detection between modalities, on which McNemar’s tests were conducted are shown in Appendix A.

### 3.2. Region Analysis

Region analysis was performed in the following anatomic areas: bones, lungs, mediastinum, adrenals, liver, abdomen and pelvis, and head and neck. ^68^Ga-DOTATATE PET/CT and ^18^F-FDG identified 32 of 36 (88.9%, CI 73.9–96.9) possible positive anatomic regions across the eleven patients. CT/MRI identified lesions in 30 of 36 (83.3%, CI 67.2–93.6) and ^18^F-FDOPA in 14 of 24 (58.3%, CI 36.6–77.9) anatomic regions. ^68^Ga-DOTATATE demonstrated the highest region detection rates in bone and head/neck lesions, ^18^F-FDG PET/CT was the highest in the mediastinum and abdominopelvic compartment, and CT/MRI was the highest in the lungs. There was a 100% detection rate in the adrenal regions, which consisted of two adrenal gland lesions in two patients by ^68^Ga-DOTATATE, ^18^F-FDG, and CT/MRI. ^18^F-FDG and CT/MRI were also superior in detecting liver lesions. ^18^F-FDOPA showed inferior detection rates compared to the other modalities across all anatomic regions. Detection rates in these regions by the imaging modalities are shown in Table 3.

### 3.3. Patient Analysis

On patient analysis, imaging performance found ^68^Ga-DOTATATE and ^18^F-FDG did not detect disease in one out of the eleven patients (different patient in either modality). ^18^F-FDOPA did not miss detection in any of the seven patients. CT/MRI did not detect in two of the eleven patients. Patient level results are detailed in Table 4. 

### 3.4. Summary of Patient, Region, and Lesion Analyses

The performances of imaging modalities in detection of *SDHA*-related metastatic lesions on a patient level, region level, and lesion level are summarized in Table 4.

### 3.5. Lesions outside the Reference Standard

Other lesions were noted on evaluation, which did not qualify for the reference standard using the imaging comparator. ^68^Ga-DOTATATE identified 70 additional lesions—68 in bones, 1 in the abdomen/pelvis, and 1 in the mediastinum. ^18^F-FDG identified 13 additional lesions to the imaging comparator—7 in bones, 3 in the abdomen/pelvis, 2 in the head/neck, and 1 in the mediastinum. CT/MRI identified 10 additional lesions—4 in lungs, 3 in the liver, 2 in the adrenals, and 1 in the abdomen/pelvis. ^18^F-FDOPA identified 12 additional lesions—11 in bones and 1 in the head/neck.

### 3.6. Patients with Four Imaging Modalities

Additionally, the total lesion detection in seven patients for whom all four imaging modalities were conducted found that ^68^Ga-DOTATATE identified 80 of 102 (78.4%, CI 69.2–86.0), ^18^F-FDG identified 81 of 102 (79.4%, CI 70.3–86.8), ^18^F-FDOPA identified 39 of 102 (38.2%, CI 28.8–48.4), and CT/MRI identified 55 of 102 (53.9%, CI 43.8–63.8). The results are detailed in Table 5. Detection of lesions by ^68^Ga-DOTATATE and ^18^F-FDG performed similarly in these seven patients. McNemar’s tests conducted to compare lesion detection rates found the following two-sided *p*-values: 1.00 for ^68^Ga-DOTATATE vs. ^18^F-FDG, less than 0.0001 for ^68^Ga-DOTATATE vs. ^18^F-FDOPA, and 0.001 for ^68^Ga-DOTATATE vs. CT/MRI (Contingency Tables shown in Appendix A).

An imaging comparison of ^68^Ga-DOTATATE, ^18^F-FDG, and ^18^F-FDOPA PET images is depicted in Figure 1.

## 4. Discussion

The study determines the performance of ^68^Ga-DOTATATE, ^18^F-FDG, ^18^F-FDOPA, and CT/MRI in a small cohort of patients with *SDHA*-related metastatic PPGL, which have been found to be aggressive in their clinical course in our institutional experience [8]. The study is important in the clinical care of patients since imaging can localize the disease burden which translates to the urgency in therapeutic selection and intervention. Analyzing how imaging modalities perform in detecting *SDHA-*related PPGL may translate to optimized clinical management. EANM/SNMMI outlined recommendations in 2019, prioritizing ^68^Ga-DOTATATE in metastatic *SDHx-*related PPGL, while our study corroborates these guidelines; based on our results, there are certain additional suggestions in imaging recommendations for *SDHA* patients with metastatic PPGL [13]. 

The EANM/SNMMI guidelines recommend ^68^Ga-DOTATATE as first-line for extra-adrenal sympathetic, metastatic, multifocal, and *SDHx-*related PPGLs with both ^18^F-FDG and ^18^F-FDOPA recommended as unequivocal second-line options. Our study identifies that ^68^Ga-DOTATATE displayed an overall lesion detection rate of 88.6%, where ^18^F-FDG had a rate of 82.9%, ^18^F-FDOPA of 39.8%, and CT/MRI of 58.9%. Upon evaluating detection rates among the PET modalities by anatomic regions, ^68^Ga-DOTATATE performed best in detecting bone and head/neck lesions, ^18^F-FDG was best in detecting mediastinal and abdominal/pelvic lesions, and ^18^F-FDOPA was inferior in all anatomic regions. In a head-to-head comparison, ^18^F-FDG outperformed ^68^Ga-DOTATATE in soft tissue organ lesions, apart from head and neck lesions (Table 2). The performance of ^18^F-FDG in detecting lesions of the lungs, mediastinum, liver, and abdomen/pelvis should inform clinicians to consider using this modality when *SDHA*-related metastatic disease is suspected in these regions.

To put in perspective the performance of ^18^F-FDG in the lungs, mediastinum, liver, and abdomen/pelvis, the results were compared to the performance of ^68^Ga-DOTATATE on an anatomic region basis. The ratio of regions detected by ^68^Ga-DOTATATE were 4/4 lung, 5/6 mediastinal, 2/2 liver, and 6/7 abdomen/pelvis (Table 2). Despite the better performance of ^18^F-FDG on a per lesion basis in these soft tissue regions, ^68^Ga-DOTATATE was able to detect a lesion in all regions but two—one mediastinum and one abdomen/pelvis—where there were only single lesions identified by the reference standard. Comparatively, CT/MRI region performance detected 4/4 lung, 3/6 mediastinal, 2/2 liver, and 7/7 abdomen/pelvis.

The CT fused with the PET imaging is a low-dose modality performed without contrast in this study used for the purpose of anatomic localization and attenuation correction. The fusion CT is not of diagnostic quality, unlike the CT used in the study for detection of lesions. Contrast-enhanced CT fused with PET or PET/MRI could provide an optimized and effective single diagnostic imaging modality. In lesion detection, CT/MRI identified 24 lesions which were negative on ^68^Ga-DOTATATE for a combined detection rate of 273/281 (97.2%). On ^18^F-FDG, CT/MRI identified 42 negative lesions for a combined detection rate of 275/281 (97.9%). While 11 lesions were not imaged by CT/MRI and were by ^68^Ga-DOTATATE and ^18^F-FDG, the improvement in lesion detection rates by combining the two modalities shows that contrast-enhanced PET/CT or PET/MRI could be a preferred modality in *SDHA-*related metastatic PPGL [15].

There were 32 lesions not identified by ^68^Ga-DOTATATE PET/CT compared to the reference standard. The missed lesions were distributed in the following regions across eight patients: 15 bone lesions (of which 14 were identified by ^18^F-FDG, 8 of possible 9 by ^18^F-FDOPA, and 8 by CT/MRI), 3 mediastinal (of which 3 were identified by ^18^F-FDG, 2 by ^18^F-FDOPA, and 2 by CT/MRI), 8 lung (of which 8 were identified by ^18^F-FDG, 2 of possible 6 by ^18^F-FDOPA, and 8 by CT/MRI), 3 liver (of which 3 were identified by ^18^F-FDG, 0 of possible 2 by ^18^F-FDOPA, and 3 by CT/MRI), 2 abdominopelvic (of which 1 was identified by ^18^F-FDG, 2 by ^18^F-FDOPA, and 2 by CT/MRI), and 1 head/neck (of which 1 was identified by ^18^F-FDG and 1 by CT/MRI).

Comparatively, 48 lesions were not identified by ^18^F-FDG PET/CT distributed in the following anatomic regions: 35 bone, 2 mediastinal, 4 lung, 1 abdominopelvic, and 6 head/neck across nine patients. There were 111 lesions not identified on CT/MRI and 63 lesions not identified on ^18^F-FDOPA PET/CT.

Limitations of this study include a small number of patients, given that this is a rare condition; however, it is the first and largest study in *SDHA-*related metastatic PPGLs. A limitation of targeting by ^18^F-FDG is its less specific mechanism of tumor uptake, which may identify a large variety of benign and malignant processes compared to ^68^Ga-DOTATATE and ^18^F-FDOPA. The detection of PPGLs by these PET modalities occurs through the following mechanisms: ^68^Ga-DOTATATE targets overexpressed somatostatin transmembrane receptors (SSTRs), ^18^F-FDOPA enters cells through large amino acid transport channels (LAT), and ^18^F-FDG enters cells via glucose transporters (GLUTs) [16]. ^18^F-FDG rapidly enters cells using GLUTs in which high rates of metabolism are occurring, such as the cerebrum, myocardium, cancers, and inflammatory processes [17]. Therefore, the low specificity and high sensitivity of ^18^F-FDG present a possibility of identifying processes unrelated to *SDHA*-related PPGL.

Patients with metastatic PPGL bone lesions have less morbidity compared to metastatic lesions in liver, retroperitoneum, and lungs [18,19,20,21]. The approach for therapy, in turn, may differ for exclusively osseous metastases versus soft tissue metastases. Locoregional therapies, such as surgery and radiation, and systemic therapies, such as chemotherapy, peptide receptor radionuclide therapy (PRRT), and receptor analogs, are all potential treatment options in any type of metastases. In addition to these approaches, patients with PPGL-related bone metastases may also benefit from therapies such as bisphosphonates or a RANKL inhibitor [22]. 

In our cohort of *SDHA* patients, bone metastases were the most abundant lesions compared to all other types of metastases combined, which is consistent with findings in all PPGL-related metastatic disease [23]. ^68^Ga-DOTATATE outperformed the other imaging modalities at detecting bone metastases, which compromised 223 of the 281 potential lesions. The sizes of these bone metastases were not measured on PET/CT, due to the difficulty of delineating tumor margins on a low-dose non-contrast fusion CT. However, the smaller sizes of numerous bone lesion may contribute to the detection differences we observed between functional and anatomic imaging. From the perspective of identifying more aggressive lesions than bone, ^18^F-FDG detected more in the lungs and liver, supporting that ^18^F-FDG may be utilized in combination with ^68^Ga-DOTATATE for this cohort of patients, rather than a second-line modality.

In terms of detecting disease on a patient-by-patient basis, both ^68^Ga-DOTATATE and ^18^F-FDG did not detect a lesion in one patient (different patient in each modality). Patient 4 had disease that was not detected by initial ^68^Ga-DOTATATE imaging; however, in subsequent annual follow-up scans, the abdominal lesion and bone lesions were avid on the modality. From a clinical viewpoint, detection on ^68^Ga-DOTATATE in any *SDHA-*related PPGL metastases translates to the opportunity of using PRRT to target overexpressed transmembrane receptors on tumors. 

Advanced metastatic PPGL is uncurable, leading patients and clinicians to seek therapies that stabilize disease, minimize progression, and reduce symptoms. ^177^Lu-DOTATATE (Lutathera^®^), which is approved in other NETs, awaits phase II trial (NCT03206060) results in PPGL, offering an approved treatment option in ^68^Ga-DOTATATE-avid PPGL [24]. In addition to ^177^Lutetium (^177^Lu), patients with SSTR-avid disease can be considered for ^90^Yttrium (^90^Y)-based PRRTs [25]. In our cohort, Patients 8 and 10 received Lutathera^®^, Patient 5 received ^90^Y-DOTATOC, and Patient 3 received ^90^Y-DOTATOC followed by ^177^Lu-DOTATOC. Patient 3 had a rapidly deteriorating clinical course in which lesions erupted after PRRT [26].

In our analysis, the results could not determine the impact of advanced metastatic PPGL and somatostatin receptor differentiation on the tumors. In the four deceased patients with high Ki-67 proliferative indices (greater than 10%), ^68^Ga-DOTATATE performed better in Patients 3 and 8, while ^18^F-FDG performed better in Patients 9 and 10. 

Larger studies are difficult to organize in an *SDHA* cohort of PPGL patients due to the low occurrence and penetrance rates in the population [12,27,28]. A multicenter approach to accumulate a sizable cohort of *SDHA*-related patients could support these findings and help draw more accurate conclusions on the role of each imaging modality in these patients.

## 5. Conclusions

In conclusion, our study in *SDHA*-related metastatic PPGL supports current guidelines recommending ^68^Ga-DOTATATE as the agent of choice, as well as the use of ^18^F-FDG as a second-line agent. In addition, we show that for *SDHA* patients with extensive lesions in the soft tissue regions (i.e., lungs, mediastinum, liver, and abdomen/pelvis), a dual approach of ^68^Ga-DOTATATE and ^18^F-FDG should be utilized for maximal sensitivity in evaluation of disease. Of note, in contrast with recommended guidelines, we show that ^18^F-FDOPA performs poorly in *SDHA* patients and should not be relied upon in this subgroup. Early detection of *SDHA*-related PPGL benefits the patient by leading to individualized management and therapy selection, which may reduce morbidity related to disease. Clinical decisions and the care of patients are often guided by imaging, especially in the case of ^68^Ga-DOTATATE, which can help clinicians identify whether PRRT is the appropriate option for the patient.

## Figures and Tables

**Figure 1 cancers-14-03886-f001:**
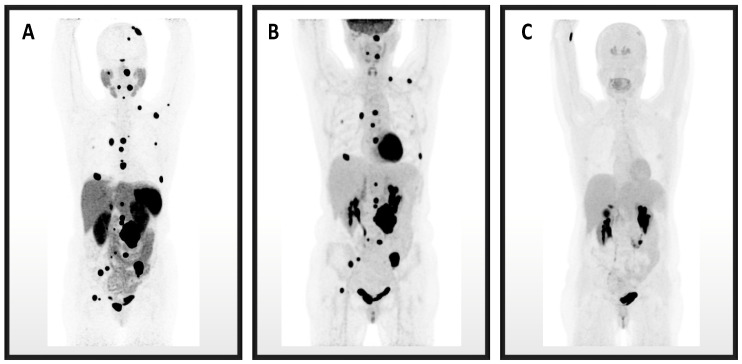
These are maximum intensity projection image PET/CTs for Patient 3. (**A**) ^68^Ga-DOTATATE identified the most PPGL lesions, followed by (**B**) ^18^F-FDG, and the fewest lesions were identified on (**C**) ^18^F-FDOPA.

**Table 1 cancers-14-03886-t001:** Clinical Characteristics of *SDHA* Patients in the study.

PT ID ^i^	Sex	SDHA Mutation	Age (d) ^ii^	Age (s) ^iii^	Primary Tumor	Biochemical Phenotype/s ^iv^	Time to Metastasis	Metastatic Location/s	Treatment/s	Ki-67 ^v^	Deceased
1	f	c.91C>T (p.Arg31*)	11	23	Left Vagale PGL	None	meta, 12 mo	Bones, Neck, Abdomen, Lung	Surgery of Primary, Surgery of Recurrence	Not available	No
2	m	c.91C>T (p.Arg31*)	57	61	Paraaortic PGL and Left Carotid Body PGL	ADR, DA	meta, 7 mo	Neck and Mediastinum	Surgery	Not available	No
3	f	c.1534C>T (p.Arg512*)	53	63	Paraaortic PGL	NA, DA	syn	Bone, Neck, Mediastinum, Abdomen, Pelvis	Partial resection of Primary; SSA; ^90^Y-DOTATOC; ^177^Lu-DOTATOC; CVD; bortezomib and clofarabine; combination capecitabine and TMZ	15% in focal areas of periaortic PGL	Yes
4	m	c.91C>T (p.Arg31*)	20	23	Aortocaval PGL	None	syn, 2 mo ^vi^	Bones and Abdomen	Surgery	Not available	No
5	m	c.91C>T (p.Arg31*)	14	16	Paracaval PGL	ADR, NA	syn	Bone	Surgery of Primary, ^90^Y-DOTATOC, SSA, ONC201	3.5% in PGL biopsy	No
6	m	c.1334C>T (p.S445L) *VUS*	53	59	Mediastinal PGL	NA, DA	meta, 48 mo	Bones and Mediastinum	Surgery, SSA, TMZ	15–20% in T10 met	No
7	m	c.91C>T (p.Arg31*)	56	67	Left Adrenal PHEO	ADR, NA, DA	meta, 120 mo	Bones, Lung, Liver, Neck	Surgery of Primary, EBRT, ^123^I-MIBG, CVD	Not available	Yes
8	f	5′UTR_3′ UTRdel	29	33	Porta Hepatis/Right Adrenal PPGL	NA	meta, 20 mo	Bones, Mediastinum, Lungs	Surgery of Primary, Surgery of Recurrence, EBRT, ^177^Lu-DOTATATE	20–30% in T7 epidural Met	Yes
9	m	c.91C>T (p.Arg31*)	44	45	Paraaortic PGL	None	syn	Bones, Lung, Mediastinum, Abdomen	Surgical decompression of Spine Met, ^123^I-MIBG, TMZ	10–15% Primary	Yes
10	f	c.91C>T (p.Arg31*)	46	54	Aortocaval PGL	ADR, NA	meta, 78 mo	Bone, Mediastinum, Liver, Abdomen, Pelvis, Neck	Surgery of Primary, EBRT, ^177^Lu-DOTATATE, CVD, Liver embolization, Liver trisegmentectomy	>20% in Liver Mets	Yes
11	f	c.91C>T (p.Arg31*)	16	17	Mediastinal PGL	None	meta, 7 mo	Bone	Resection of Primary, resection of recurrent bed	Not available	No

abbreviations: f—female, m—male, PHEO—pheochromocytoma, PGL—paraganglioma, ADR—adrenergic, NA—Noradrenergic, DA—Dopaminergic, meta—metachronous, syn—synchronous, mo—months, SSA—somatostatin analogs, TMZ—temozolomide, VUS—variant of unknown significance, EBRT—external beam radiation therapy, CVD—chemotherapeutic regimen cyclophosphamide-vincristine-dacarbazine. ^i^ PT ID is the patient identification number in the cohort. ^ii^ Age (d) is Age of PPGL diagnosis in years. ^iii^ Age (s) is Age at time of imaging scans in years. ^iv^ Biochemical elevation at time of Scans, with adrenergic, noradrenergic, and dopaminergic referring to elevations in epinephrine and/or metanephrine, norepinephrine and/or normetanephrine, and dopamine and/or 3-methoxytyramine, respectively. ^v^ Ki-67 is the cellular proliferative index staining on histopathology. ^vi^ Patient has synchronous metastases and did not have full body imaging until seen in our study.

**Table 2 cancers-14-03886-t002:** Lesion analysis displaying detection rates and confidence intervals by imaging modalities categorized by total, primary, and metastatic lesions in 11 patients. Further subcategorization of which region the primary and metastatic lesions were located, and their detection rates are shown.

	^68^Ga-DOTATATE	^18^F-FDG	^18^F-FDOPA	CT/MRI
Total Lesions	249/281 88.6 (84.3–92.1)	233/281 82.9 (78.0–87.1)	39/102 39.8 (30.2–50.2)	157/270 58.2 (52.0–64.1)
Primary Lesions	9/10 90.0 (55.5–99.8)	9/10 90.0 (55.5–99.8)	3/7 42.9 (9.9–81.6)	7/10 70.0 (34.8–93.3)
Mediastinum	1/1	1/1	- ^vii^	0/1
Adrenal	1/1	1/1	0/1	1/1
Abdomen/Pelvis	5/6	6/6	2/5	6/6
Head/Neck	2/2	½	1/1	0/2
Metastatic Lesions	240/271 88.6 (84.2–92.1)	225/271 83.0 (78.0–87.3)	36/95 37.9 (28.1–48.4)	150/260 57.7 (51.4–63.8)
Bone	208/223	188/223	27/64	110/212
Lungs	11/19	15/19	3/10	19/19
Mediastinum	4/7	5/7	3/7	4/7
Adrenal	1/1	1/1	0/1	1/1
Liver	4/7	7/7	0/4	7/7
Abdomen/Pelvis	6/7	6/7	2/5	4/7
Head/Neck	6/7	3/7	¼	5/7

^vii^ Patient did not receive ^18^F-FDOPA Imaging due logistics of scheduling.

**Table 3 cancers-14-03886-t003:** Imaging modality detection of a positive lesion in all anatomic regions and individual anatomic regions.

	^68^Ga-DOTATATE	^18^F-FDG	^18^F-FDOPA	CT/MRI
All Regions	32/36	32/36	14/24	30/36
Bone	9/10	10/10	5/6	8/10
Lungs	4/4	4/4	1/2	4/4
Mediastinum	5/6	4/6	3/5	3/6
Liver	2/2	2/2	0/1	2/2
Adrenal	2/2	2/2	0/2	2/2
Abdomen/Pelvis	6/7	7/7	3/5	7/7
Head/Neck	4/5	3/5	2/3	4/5

**Table 4 cancers-14-03886-t004:** Ratios, detection rates (DR), and confidence intervals (CI) of imaging modalities at patient level, lesion level, and region level.

	^68^Ga-DOTATATE		^18^F-FDG		^18^F-FDOPA		CT/MRI	
	Ratio	DR (CI)	Ratio	DR (CI)	Ratio	DR (CI)	Ratio	DR (CI)
Patient Level	10/11	90.9% (58.7–99.8)	10/11	90.9% (58.7–99.8)	7/7	100% (59.0–100.0)	9/11	81.8% (48.2–97.7)
Region Level	32/36	88.9% (73.9–96.9)	32/36	88.9% (73.9–96.9)	14/24	58.3% (36.6–77.9)	30/36	83.3% (67.2–93.6)
Lesion Level	249/281	88.6% (84.3–92.1)	233/281	82.9% (78.0–87.1)	39/98	39.8% (30.2–50.2)	159/270	58.9% (52.8–64.8)

**Table 5 cancers-14-03886-t005:** Lesion analysis in patients in which all four imaging modalities were performed displaying detection rates and confidence intervals in total *SDHA* lesions, primary *SDHA* lesions, and metastatic *SDHA* lesions.

	^68^Ga-DOTATATE	^18^F-FDG	^18^F-FDOPA	CT/MRI
Total Lesions	80/102 78.4 (69.2–90.0)	81/102 79.4 (70.3–86.8)	39/102 39.8 (30.2–50.2)	55/102 53.9 (43.8–63.8)
Primary Lesions	6/7 85.7 (42.1–99.6)	6/7 85.7 (42.1–99.6)	3/7 42.9 (9.9–81.6)	6/7 85.7 (42.1–99.6)
Metastatic Lesions	74/95 77.9 (68.2–85.8)	75/95 79.0 (69.4–86.6)	36/95 37.9 (28.1–48.4)	49/95 51.6 (41.1–62.0)

## Data Availability

The data presented in this study are available on request from the corresponding author. The data are not publicly available due to the protection of patient identification.

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
