# Peer review of "Performances of Functional and Anatomic Imaging Modalities in Succinate Dehydrogenase A-Related Metastatic Pheochromocytoma and Paraganglioma"

_cancers, 2022, doi:10.3390/cancers14163886_

Round 1

Reviewer 1 Report

Dear authors, regarding discussion and conclusion you did a good job, but I have to suggest some corrections. You ‘ll find all suggestions listed point by point:

Minor revisions:

- Table 1: “PHEO”, “f” “m” expansions is missing

- lines 163 and 165: I would specify better the meaning of “AL” and “JAC”

- line 202: it is recommended to explain why in patients 1, 5 and 8 you detected primary lesions despite surgery of primary.

- Table 4: I suggest to modify the graphic (under “ratio DR ratio” there is a second line that I don’t understand)

- line 301: the correct word is “Table 3” not “Table 2”

Major revisions

- lines 80-85: it is recommended to add percentage of patient with SDHx-related PPGL with appropriate references.

- line 123: it is recommended to add, not only in the table but also in the text, mutations found and if one of these is more frequently than the others.

- Paragraphs 3.3. 3.4, 3.5: I suggest to combine these paragraphs

Author Response

Thank you for your review. Please see the attachment of our responses.

Reviewer 2 Report

The Authors conducted a very sound study to identifywhich imaging modality among PET, CT, and MRI performs better in localizing metastatic PPGL lesions related to SDHA.

I feel that the results are relevant mainly for a niche of clinicians involved in the treatment of PPGL; however, the study was very well conducted and the paper is well written.

I can offer some suggestions:

- Table 1 is hard to read, can the Authors rework it to enhance clarity?

- the first part of the Discussion looks like a repetition of the Results. can the Authors consider redrafting?

- can the Authors expand on the relevance of their findings for clinical practice (see for instance lines 367-368). This would further justify publication in a general journal like Cancers.

Author Response

Thank you. Please see the attachment for our responses. 
